# Effects of Anemoside B4 on Plasma Metabolites in Cows with Clinical Mastitis

**DOI:** 10.3390/vetsci10070437

**Published:** 2023-07-05

**Authors:** Liuhong Shen, Yu Shen, Yue Zhang, Suizhong Cao, Shumin Yu, Xiaolan Zong, Zhetong Su

**Affiliations:** 1The Medical Research Center for Cow Disease, College of Veterinary Medicine, Sichuan Agricultural University, Chengdu 611130, China; xy15031324924@163.com (Y.S.); zhangyuesicau@163.com (Y.Z.); suizhongcao@126.com (S.C.); yayushumin@sicau.edu.cn (S.Y.); zongxl@sicau.edu.cn (X.Z.); 2Guangxi Innovates Medical Technology Co., Ltd., Lipu 546600, China; shayugege@126.com

**Keywords:** metabolite, clinical mastitis, dairy cows, anemoside B4

## Abstract

**Simple Summary:**

Clinical mastitis (CM) is one of the most prevalent diseases in cows. Anemoside B4, the main active ingredient in *P*. *chinensis*, has been shown to reduce the concentration of pro-inflammatory cytokines and is effective in cows with CM. However, the changes in the metabolic profile of cows with CM after treatment with anemoside B4 remain unclear. This study aimed to investigate how anemoside B4 affects the plasma metabolic profile in cows with CM and to provide a theoretical basis for its effects in veterinary practice.

**Abstract:**

Anemoside B4 has a good curative effect on cows with CM; however, its impact on their metabolic profiles is unclear. Based on similar somatic cell counts and clinical symptoms, nine healthy dairy cows and nine cows with CM were selected, respectively. Blood samples were collected from cows with mastitis on the day of diagnosis. Cows with mastitis were injected with anemoside B4 (0.05 mL/kg, once daily) for three consecutive days, and healthy cows were injected with the same volume of normal saline. Subsequently, blood samples were collected. The plasma metabolic profiles were analyzed using untargeted mass spectrometry, and the concentrations of interleukin (IL)-1β, IL-6, and tumor necrosis factor-α (TNF-α) in serum were evaluated via ELISA. The cows with CM showed increased concentrations of IL-1β, IL-6, and TNF-α (*p* < 0.05). After treatment with anemoside B4, the concentrations of IL-1β, IL-6, and TNF-α were significantly decreased (*p* < 0.01). Untargeted metabolomics analysis showed that choline, glycocholic acid, PC (18:0/18:1), 20-HETE, PGF3α, and oleic acid were upregulated in cows with CM. After treatment with anemoside B4, the concentrations of PC (16:0/16:0), PC (18:0/18:1), linoleic acid, eicosapentaenoic acid, phosphorylcholine, and glycerophosphocholine were downregulated, while the LysoPC (14:0), LysoPC (18:0), LysoPC (18:1), and cis-9-palmitoleic acid were upregulated. This study indicated that anemoside B4 alleviated the inflammatory response in cows with CM mainly by regulating lipid metabolism.

## 1. Introduction

Clinical mastitis (CM) is one of the diseases with the highest incidence rate and severity in the periparturient period of dairy cows [1]. Mastitis results in reduced milk production, impaired health among cows, and substantial economic loss every year [2]. Cows with CM have marked changes in both the appearance of their mammary gland [3] and milk consistency (clumpy, watery, bloody, or yellowish milk) [4]. Mastitis is mainly caused by intramammary bacterial infection, leading to a significant increase in the concentrations of inflammatory cytokines, including IL-1β, IL-6, and TNF-α, in cows with CM [5,6]. The ability of cows to resist CM is to some extent related to the effectiveness of the inflammatory response [7]. Chronic mammary inflammation may lead to extensive damage of the mammary gland [8]. Antibiotics are the most common treatment for CM on dairy farms [9]. However, the use of antibiotics may lead to antibiotic resistance among the pathogens responsible for clinical mastitis [10]. Antibiotic residues in milk from dairy cows also pose a significant threat to human health [11]. Many traditional Chinese medicines and their extracts have been developed into medicines for the treatment of such diseases and have the potential to address both problems [12].

*Pulsatilla chinensis* (Bge.) Reg (*Pulsatilla*), a well-known traditional Chinese herb, is commonly used to treat inflammatory diseases [13]. Previous studies have shown that the triterpenoid glycoside compounds in this herb play an important role in anti-inflammation due to their anti-microbial properties [14]. Anemoside B4, the main active ingredient in P. *chinensis*, has been shown to reduce pro-inflammatory cytokine concentrations and nuclear factor-κ-gene binding activity in kidney and lung diseases [15]. Moreover, its antioxidant and anti-inflammatory activities have been confirmed in both in vivo and in vitro experiments [16]. Our previous studies confirmed that anemoside B4 is effective for cows with CM, with a clinical cure rate of 75%; the negative conversion rate of bacteria was 69.23% for Mycoplasma, 33.33% for Coccus (including *Staphylococcus epidermidis*, *Staphylococcus chromogenes* and *Staphylococcus haemolyticus*), and 100% for *Escherichia coli* [17,18]. However, the changes in the metabolic profile of cows with CM after treatment with anemoside B4 remain unclear.

Metabolomics is useful for analyzing known metabolic pathways and functional biological changes in physiological and pathological responses [19]. It provides detailed molecular profiles of biospecimens, which can be used to obtain underlying biological information [20]. Studies have shown that plasma amino acids and sphingolipids can be used to predict risks among cows during the transition period, thereby allowing one to gain insight into the etiopathology of the disease [21]. Isoleucine and lactate are significantly increased in milk when there is a higher somatic cell count (SCC) [22]. Overall, metabolomics can be used for studying the pathology of diseases as well as for screening the biomarkers of disease in cows. In the present study, we used ultra-high-performance liquid chromatography coupled with quadrupole time-of-flight mass spectrometry (UHPLC-Q-TOF-MS) to investigate the metabolic profiles in the plasma of healthy and anemoside B4-treated cows with clinical mastitis to explore the potential role of differential metabolites in the therapeutic activity of this herb. This study aimed to investigate how anemoside B4 affects the metabolic profiles of cows with CM and to provide a theoretical basis for its effects in veterinary practice.

## 2. Materials and Methods

### 2.1. Animal Care

The collection of samples was performed in accordance with the guidelines of the Care and Use of Laboratory Animals of China (No.2013-028).

### 2.2. Preparation of Anemoside B4

The molecular formula of anemoside B4 is C_59_H_96_O_26_, the relative molecular mass is 1221.38, and the purity is 66.67% (Guangxi innovates medical technology Co., Ltd., Lipu, Guangxi, China).

### 2.3. Experimental Design, Cows, and Management

The cows examined for this study were from a larger sample of 100 cows housed in an open farm in Sichuan, China, were fed the same total mixed rations (TMR), and had unlimited access to fresh water. TMR information is presented in Appendix A. Cows with CM were diagnosed by a veterinarian based on obvious clinical signs, including udder redness, milk discoloration, clots, and SCCs in the milk exceeding 500,000 cells/mL, and had their diagnosis confirmed after we conducted laboratory examinations on pathogenic bacteria. More details can be found in our previous study [18]. A total of 12 Holstein cows were diagnosed with clinical mastitis (milk SCCs = 2,844,100 ± 61.64 cells/mL; milk yield = 25.68 ± 2.13 kg/day; parity = 2.65 ± 0.27). These cows were given anemoside B4 (0.05 mL/kg, once daily) via intramuscular injection for three consecutive days (cows had not received any antibiotic treatment before this study). After treatment, the symptoms of 9 cows were relieved and SCCs < 200,000 cells/mL. They were assigned to the experimental group. Nine disease-free dairy cows from the same breed and with similar body conditions (milk SCCs = 184,700 ± 4.13 cells/mL; milk yield = 37.97 ± 3.37 kg/day; parity 2.48 ± 0.34; free of diseases) were selected as controls. These cows received an injection of the same volume of normal saline as anemoside B4 in mastitis cows.

### 2.4. Sample Collection

Blood samples of cows in both groups were collected from the caudal vein on the day of CM diagnosis (CM-0). After 3 consecutive days of anemoside B4 injection (CM-3), blood samples were collected again. Plasma was collected in two different ways: one was collected using heparin sodium as an anticoagulant and centrifuged at 1500× *g* for 10 min at room temperature, and the other was collected without an anticoagulant and then centrifuged under the same conditions. All samples were stored at −80 °C.

### 2.5. Biochemical Analysis

The serum concentrations of interferon-1β (IL-1β), interferon-6 (IL-6), and tumor necrosis factor-α (TNF-α) were measured using ELISA kits (Enzyme-linked Biotechnology Co., Ltd. Shanghai, China) according to the manufacturer’s instructions.

### 2.6. Metabolic Profiling Analysis

Plasma samples were separated via UHPLC (1290 infinite LC, Agilent Technologies, Santa Clara, CA, USA) using a HILIC column with 25 °C column temperature and 0.3 mL/min velocity. The mobile phase was A: 25 mM ammonium acetate and 25 mM ammonia in water, B: acetonitrile. The gradient was 85% B for 1 min, linearly reduced to 65% in 11 min, then reduced to 40% in 0.1 min and kept for 4 min, and then increased to 85% in 0.1 min, with a 5 min re-equilibration period being employed.

The AB triple TOF 6600 mass spectrometer was used to obtain the samples’ primary and secondary spectra. The ESI source conditions after HILIC chromatographic separation were as follows: Ion Source Gas1: 60, Ion Source Gas2: 60, Curtain gas: 30, source temperature: 600 °C, Ion Spray Voltage Floating ± 5500 V, TOM MS scan *m*/*z* range: 60–1000 Da, product ion scan *m*/*z* range: 25–1000 Da, TOF MS scan accumulation time 0.20 s/spectra, product ion scan accumulation time: 0.05 s/spectra, the secondary mass spectrum was obtained by information-dependent acquisition (IDA) with high-sensitivity mode selected, declustering potential (DP): ±60 V, collision energy was fixed at 35 V ± 15 eV; IDA was set as follows: exclude isotopes within 4 Da, Candidate ions to monitor per cycle: 10.

### 2.7. Data Processing and Statistical Analysis

Raw data were converted into mzXML files using ProteoWizard MS Convert. Peak alignment, retention time correction, and peak area extraction were completed used R package XCMS. More than 50% of the non-zero measurement values in at least one group were maintained. The data were pre-processed via pareto-scaling, after which pattern recognition was performed using the SIMCA-P software. The metabolites were identified via accuracy mass (<25 ppm).

Analysis of differences between groups (H vs. CM-0 and CM-0 vs. CM-3) was per-formed using principal component analysis (PCA) and orthogonal partial least square discriminant analysis (OPLS-DA). Multivariate data analysis was performed in R software to evaluate the robustness of the model. Significantly different metabolites were screened out, with variable influence on projection (VIP) values > 1.0 and *p*-values < 0.05 for the raw data. Differential metabolites were further identified and analyzed using the KEGG online database (https://www.kegg.jp/kegg/pathway.html (accessed on 20 December 2022)) and MetaboAnalyst 5.0 (https://www.metaboanalyst.ca/ (accessed on 20 December 2022)).

Differences in serum biochemical parameters were analyzed using Student’s *t*-test (SPSS version 24.0). Data are presented as mean ± SEM. Differences were considered significant at *p* < 0.05 and highly significant at *p* < 0.01. Graphs were created using GraphPad Prism 8.0.

## 3. Results

### 3.1. Biochemical Analysis

As shown in Figure 1, the serum concentrations of IL-1β were significantly increased in CM cows (*p* = 0.019) before anemoside B4 treatment, as were the concentrations of IL-6 and TNF-α (*p* < 0.01), compared with the healthy cows. After anemoside B4 treatment, the IL-1β, IL-6, and TNF-α concentrations were significantly reduced (*p* < 0.01) in cows with CM compared with CM-0 cows.

### 3.2. Metabolite Profiles and Data Analysis

Our results showed that the retention time, peak, and intensity of each sample shared a very high similarity, indicating that the system was stable throughout the experimental process (Appendix A). A PCA showed that the three groups could cluster, and the healthy group could be separated with the group CM-0 and group CM-3, respectively. There were no significantly differences between group CM-0 and group CM-3 (Figure 2a,b). Hence, OPLS-DA was performed. When the data from the healthy group were compared with that of the CM-0 group, the OPLS-DA results showed clear differences in the positive and negative modes. The OPLS-DA score plot in both the positive and negative modes was R^2^Y = 0.966, Q^2^ = 0.663 and R^2^Y = 0.992, Q^2^ = 0.586, respectively (Appendix A). The CM-0 groups and CM-3 groups showed notable differences in each ion mode. The parameters were R^2^Y = 0.991, Q^2^ = 0.445 and R^2^Y = 0.968, Q^2^ = 0.196 in the positive and negative modes (Appendix A). The parameters of OPLS-DA were close to 1, indicating that there was a notable difference in metabolite levels between each group [23]. The permutation testing of groups in each ion mode revealed that the observed separation was not by chance (*p* < 0.01) and that the cross-validation results were reliable, indicating a lack of overfitting of the model. Hence, the differential metabolites can be identified according to it.

### 3.3. Differential Metabolites among Groups

We identified 41 metabolites that differed between the H and CM-0 groups (VIP score > 1 and *p* < 0.05) (Appendix A). These metabolites were mainly amino acids (AAs), nucleotides, and lipids. Twenty metabolites differed between the CM-0 and CM-3 groups (Appendix A), which were mainly lipids. Furthermore, all metabolites that differed among the groups were analyzed. The fold change bar (Figure 3) shows the changes in metabolites among the three groups. Five lipids and one bile acid in the CM-0 group were increased: choline (VIP = 1.75, *p* < 0.05), PC (18:0/18:1) (VIP = 3.76, *p* < 0.01), oleic acid (VIP = 4.49, *p* < 0.05), PGF3α (VIP = 1.03, *p* < 0.05), 20-hydroxyeicosatetraenoic acid (20-HETE) (VIP = 1.32, *p* < 0.05), and glycocholic acid (VIP = 2.21, *p* < 0.01). Conversely, AAs and its metabolites and nucleotides were decreased in cows with CM (Figure 3a). After anemoside B4 treatment (Figure 3b), PC (16:0/16:0) (VIP = 1.59, *p* < 0.01), PC (18:0/18:1) (VIP = 5.17, *p* < 0.05), linoleic acid (VIP = 1.12, *p* < 0.05), phosphorylcholine (VIP = 3.19, *p* < 0.05), eicosapentaenoic acid (EPA) (VIP = 2.74, *p* < 0.05), and glycerophosphocholine (VIP = 2.53, *p* < 0.05) were decreased in cows with CM. Conversely, LysoPC (18:0) (VIP = 6.22, *p* < 0.05), LysoPC (18:1) (VIP = 6.15, *p* < 0.01), LysoPC (14:0) (VIP = 4.44, *p* < 0.05), and cis-9-palmitoleic acid (VIP = 4.91, *p* < 0.05) were increased.

### 3.4. Pathway Analysis

The total differential metabolites among the three groups were enriched in two KEGG pathway hierarchies, respectively. Most metabolites were involved in pyrimidine metabolism; aminoacyl-tRNA biosynthesis; valine, leucine, and isoleucine biosynthesis; and pantothenate and CoA biosynthesis. There was a significant difference in arginine biosynthesis between the H and CM-0 groups (Figure 4a). After treatment with anemoside B4, the changes in metabolites were mainly observed in the biological pathways, including glycerophospholipid metabolism, linoleic acid metabolism, and the biosynthesis of unsaturated fatty acids (Figure 4b). To further determine the biological significance of anemoside B4 in mastitis cows, we pooled the metabolites to provide a summary of complex processes (Figure 4c).

## 4. Discussion

Studies have shown that anemoside B4 alleviates lipopolysaccharide (LPS) triggered acute inflammatory responses, resulting in a reduction in pro-inflammatory cytokines [15,16]. Our previous study also showed that anemoside B4 has positive effects on cows with CM; anemoside B4 was effective against *Staphylococcus epidermidis*, *Staphylococcus chromogenes*, *Staphylococcus haemolyticus*, and *Escherichia coli* in cows with CM, but was not effective against *Staphylococcus aureus* [18]. This study showed that anemoside B4 can significantly reduce inflammatory factors in cows with mastitis, which was mutually confirmed by the results of a previous clinical efficacy study [18]. On the basis of these studies, we explored the changes regarding plasma metabolites in cows with CM before and after treatment with anemoside B4. The three cows that passed away most likely had CM caused by *Staphylococcus aureus*. Therefore, we performed a subsequent metabolomics analyses only on the cured cows.

During mastitis, the degree of inflammation in the mammary glands continues to develop. The results of our study are in line with the phenomenon, showing that the concentrations of inflammatory cytokines (IL-1β, IL-6 and TNF-α) increased in cows with CM. This phenomenon was then reversed via anemoside B4 treatment. With a decrease in concentrations of IL-1β, IL-6, and TNF-α, the plasma metabolites also changed. Here, we found that glycocholic acid, a bile acid, was significantly increased in cows with CM. During human pregnancies, pregnancy hormones slow down bile flow, resulting in bile acids spilling into the blood stream [24]. A previous study indicated that the bile acids combined with the cell membranes to release arachidonic acid, further triggering DNA damage [25]. These all supported our observation that there was higher glycocholic acid concentrations in the cows with CM. The concentrations of PC (18:0/18:1), oleic acid, PGF3α, and 20-HETE were increased in the cows with CM. 20-HETE is a metabolic product of arachidonic acid (ARA), which correlates with the acute inflammatory phase of CM in cows [26,27]. PGF3α is a bioactive EPA metabolite synthesized via the cyclooxygenase pathway [28]. The elevation of PGF3α has a positive effect on CM cows, and oleic acid plays an important role in membrane structure; an increase in both is associated with an increased risk of chronic diseases in humans [29]. Choline provides methyl groups for carnitine biosynthesis, which could link the water-soluble phospholipids, PC, LysoPC, phosphorylcholine, and glycerophosphocholine with AA and its metabolites [30]. It has also been reported that various diseases are associated with elevated circulating choline concentrations [31]. Thus, it could be said that the lipid mediators were signaling molecules that modulated inflammation.

After anemoside B4 treatment, most metabolites were mainly involved in the following three metabolic processes in mastitis-infected cows: glycerophospholipid metabolism, linoleic acid metabolism, and the biosynthesis of unsaturated fatty acids. Among these, linoleic acid is an omega-6 unsaturated fatty acid with pro-inflammatory effects and is reported to be increased in the plasma of cows with CM [27]. However, we observed a decrease in linoleic acid in the cows with CM after anemoside B4 treatment. The present study showed that the concentrations of EPA were downregulated after treatment with anemoside B4. A previous study showed that decreases in IL-1 and TNF were accompanied by a decreased ARA/EPA ratio during parturition [32]. Similarly, the downregulation of EPA might be accompanied by reduced inflammation in cows with CM. As mentioned above, phosphorylcholine and glycerophosphocholine were decreased in the cows with CM after anemoside B4 treatment, which also was accompanied by reduced IL-1β, IL-6, and TNF-α. Overall, our results indicate that anemoside B4 could inhibit the development of mastitis and improve the recovery of cows with CM. Interestingly, contradictory changes were observed in PC (18:0/18:1) and LysoPC (14:0) before and after anemoside B4 administration. PC metabolism is enhanced under inflammatory conditions (accompanied by increasing ARA and linoleic acid production) [33]. The concentrations of members of the LysoPC family, including LysoPC (14:0), LysoPC (18:0), and LysoPC (18:1,) were increased in cows with CM undergoing treatment anemoside B4 with. LysoPC is generated by the enzyme phospholipase A2, which hydrolyses PC at the sn-2 position [34]. The LysoPC/PC ratio has been proposed as an inflammation marker; for example, a lower LysoPC/PC ratio is associated with an increased risk of mortality among patients with sepsis [35]. In the study conducted by the authors of [36], a decrease in the LysoPC/PC ratio was accompanied by an acute CM infection in dairy cows because pathogenic bacteria alter phospholipid metabolism [36]. This study suggests that LysoPC might be involved in the progression and carcinogenesis of CM, as shown in the present study. Moreover, anemoside B4 can coordinate lipid metabolism, thereby alleviating clinical mastitis.

In this study, the concentrations of AAs and its metabolites and nucleotides and its metabolites were decreased in cows with CM. Previous studies have shown that the extravasation of blood proteins into the mammary gland through the blood mammary barrier, which possibly occurs due to consumption during a systematic inflammatory response in a negative acute phase protein response [37]. Pyrimidine nucleotides are essential for many cellular processes, and the disruption of pyrimidine metabolism has been associated with various clinical diseases [38]. However, we did not identify any changes in these metabolites before or after anemoside B4 treatment.

## 5. Conclusions

This study showed the effects of anemoside B4 on inflammatory factors and metabolites in cows with CM. At the start of the study, among the dairy cows with CM, concentrations of lipids metabolites were upregulated, while the concentrations of AA and its metabolites and nucleotides decreased. After treatment with anemoside B4, lipids, especially PC (16:0/16:0), PC (18:0/18:1), linoleic acid, phosphorylcholine, EPA, and glycerophosphocholine, were decreased in the cows with CM. Additionally, LysoPC (18:0), LysoPC (18:1), and LysoPC (14:0), as well as cis-9-palmitoleic acid, were increased. To conclude, anemoside B4 may have a positive regulatory effect on plasma metabolites in cows with CM.

## Figures and Tables

**Figure 1 vetsci-10-00437-f001:**
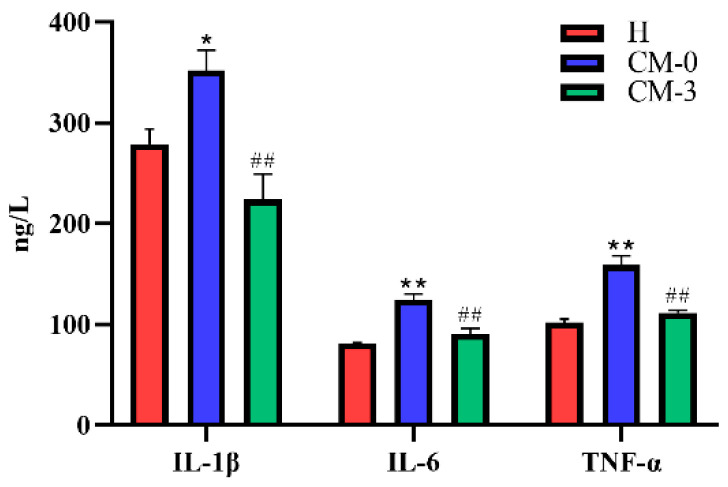
Serum concentrations of IL-1β, IL-6, and TNF-α in healthy cows and cows with CM before and after treatment with anemoside B4. H, healthy; CM, clinical mastitis; CM-0 is the day of diagnosis, CM-3 represents the three consecutive days of anemoside B4 treatment. Each column represents a group of samples. Data are presented as mean ± SEM. SEM means standard error of the mean. * *p* < 0.05; ** *p* < 0.01, CM-0 vs. H; ^##^ *p* < 0.01 CM-0 vs. CM-3.

**Figure 2 vetsci-10-00437-f002:**
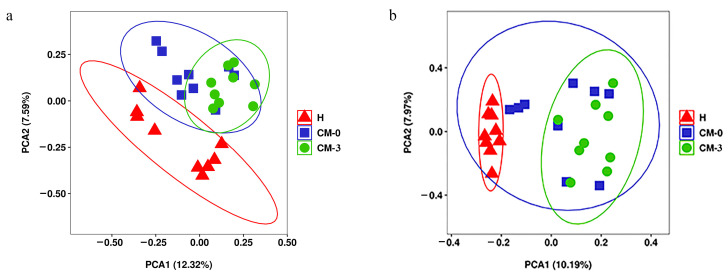
PCA score plot of the H group, CM-0 group, and CM-3 group. (**a**) Positive electrospray ionization mode (ESI+). (**b**) Negative electrospray ionization mode (ESI−). PC, principal component; H, healthy; CM, clinical mastitis; CM-0 is the day of diagnosis, CM-3 represents the three consecutive days of anemoside B4 administration.

**Figure 3 vetsci-10-00437-f003:**
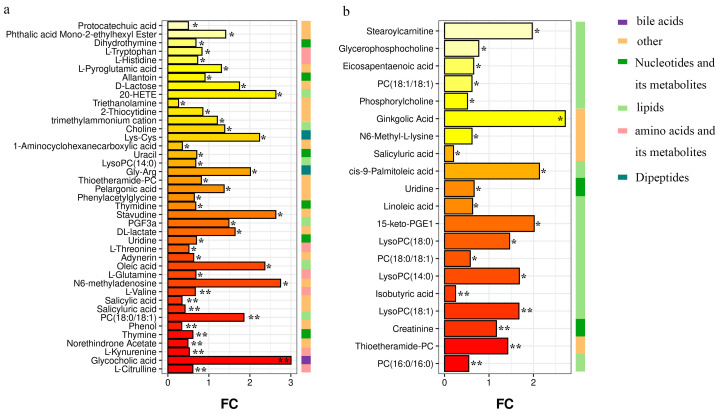
Differential metabolites analysis. (**a**,**b**) FC bar graphs of metabolites in H vs. CM-0 group and CM-0 vs. CM-3 group. The length of the bar represents the fold change (FC) of the metabolite between the two groups. The color of the bar indicates the significance of the metabolite difference between the two groups. FC > 1 stands for upregulated metabolites, FC < 1 stands for downregulated metabolites. * *p* < 0.05; ** *p* < 0.01. H, healthy; CM, clinical mastitis; CM-0 is the day of diagnosis, CM-3 represents the three consecutive days of anemoside B4 treatment.

**Figure 4 vetsci-10-00437-f004:**
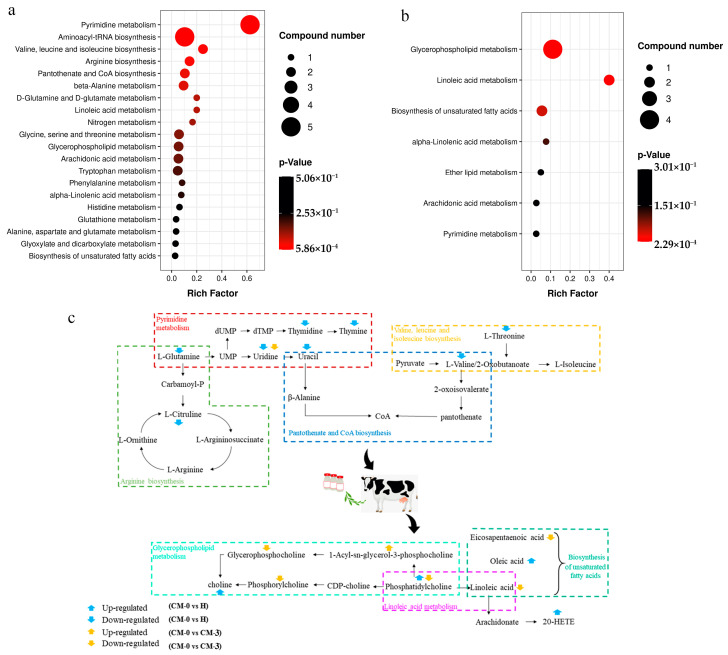
(**a**) Enrichment of differential metabolite KEGG pathway between H and CM-0 groups; (**b**) Enrichment of differential metabolite KEGG pathway between CM-0 and CM-3 groups. The x-axis denotes the rich factor. The y-axis indicates the metabolic pathway. Bubble size means the number of metabolites in the pathway; bubble color means *p*-value; the darker the color (red), the smaller the *p*-value; the darker the color (black), the higher the *p*-value. (**c**) Network metabolic map for differential metabolites. The altered pathways referred to pyrimidine metabolism, valine, leucine, and isoleucine biosynthesis, arginine biosynthesis, pantothenate, and CoA biosynthesis in cows with clinical mastitis; the targeted metabolic pathways in the CM cows treated with anemoside B4 mainly focused on glycerophospholipid metabolism, linoleic acid metabolism, and the biosynthesis of unsaturated fatty acids (*p* < 0.05). H, healthy; CM, clinical mastitis; CM-0 is the day of diagnosis, CM-3 represents the three consecutive days of anemoside B4 treatment.

## Data Availability

The data presented in this study are available in the article/Appendix A; further inquiries can be directed to the corresponding author.

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
