# Peer review of "Effects of Anemoside B4 on Plasma Metabolites in Cows with Clinical Mastitis"

_vetsci, 2023, doi:10.3390/vetsci10070437_

Round 1

Reviewer 1 Report

In this manuscript, the authors propose that studied the changes in blood metabolism profiles of healthy and mastitis cows, as well as the therapeutic mechanism of B4 on mastitis. This study is innovative to some extent, and the scientific research level meets the requirements for publication in this journal, but there are many details to be modified.

1.            Major concerns

The author needs to provide more additional information about dairy cows, as there are many studies suggested that hyperketosis, hyperFFA, or calving frequency are closely related to mastitis. Especially, ketosis or subclinical ketosis must be excluded before the experiment begins if the author wants to prove the changes in lipid levels during mastitis in dairy cows.

The manuscript is not written in clear and grammatically correct English, and requires English language editing services.

2.            Minor concerns

a. Lines73-75What is the source of Anemoside B4

b. Line155: 0,01 or 0.01?

c. The order of descriptions in the annotations should be consistent (such as Figures 1, 3, or 4)

d. Lines 257-258: The linoleic acid (LA) has been defined in Line 214, and all abbreviations should be applied in the following sections of the article, rather than being redefined in 257 or using the full name in 258.

e. The conclusion should be more comprehensive and concise.

The manuscript is not written in clear and grammatically correct English, and requires English language editing services.

Author Response

  1. Comments and Suggestions for Authors

In this manuscript, the authors propose that studied the changes in blood metabolism profiles of healthy and mastitis cows, as well as the therapeutic mechanism of B4 on mastitis. This study is innovative to some extent, and the scientific research level meets the requirements for publication in this journal, but there are many details to be modified.

  1. Major concerns

The author needs to provide more additional information about dairy cows, as there are many studies suggested that hyperketosis, hyperFFA, or calving frequency are closely related to mastitis. Especially, ketosis or subclinical ketosis must be excluded before the experiment begins if the author wants to prove the changes in lipid levels during mastitis in dairy cows.

Answer: Thanks for your suggestions and we do really agree with your opinions. The diseased animals in the present study were diagnosed with clinical mastitis based on their clinical symptoms, somatic cell count in milk and laboratory examinations of bacteria (published in previous studies). The causes of cow mastitis are complex and the main purpose of this study was to explore the changes in metabolites in clinical mastitis cows treated by anemoside B4, so the experiment was designed in this way.

The manuscript is not written in clear and grammatically correct English, and requires English language editing services.

Answer: The language of the manuscript has been extensively edited.

  1. Minor concerns
  2. Lines73-75:What is the source of Anemoside B4?

Answer: Anemoside B4 is the main active ingredient of P. chinensis, which is a traditional Chinese herb. The molecular formula of anemoside B4 is C59H96O26. The anemoside B4 used in the present study was from a company. Details have been added in the manuscript.

  1. Line155: 0,01 or 0.01?

Answer: Thanks, it should be 0.01. It has been corrected.

  1. The order of descriptions in the annotations should be consistent (such as Figures 1, 3, or 4)

Answer: It has been revised in the manuscript.

  1. Lines 257-258: The linoleic acid (LA) has been defined in Line 214, and all abbreviations should be applied in the following sections of the article, rather than being redefined in 257 or using the full name in 258.

Answer: It has been revised in the manuscript.

  1. The conclusion should be more comprehensive and concise.

Answer: Thanks, the Conclusion has been edited based on your suggestions.

Reviewer 2 Report

The authors present a manuscript "Effects of anemoside B4 on plasma metabolites in cows with clinical mastitis". This is an interesting work on a very important topic, especially in dairy animals. However, despite the work's merits, the manuscript in its present form is confusing and may be misleading for the reader.

The work is initially presented in the simple summary, abstract and introduction in the context of strategies to combat antibiotic resistance. Although, all work is directed towards the possible anti-inflammatory effects of anemoside B4.

In a study on clinical mastitis in cattle, the clinical diagnosis and SCC only establish the presumptive diagnosis. Laboratory confirmation of the agents involved is mandatory.

Thus, the manuscript must be rewritten and resubmitted. With special attention to the study framework, clear definition of objectives, objective-oriented discussion (the authors already have an adequate discussion directed towards laboratory results) and respective conclusions

Moderate editing of English language required

Author Response

Comments and Suggestions for Authors

The authors present a manuscript "Effects of anemoside B4 on plasma metabolites in cows with clinical mastitis". This is an interesting work on a very important topic, especially in dairy animals. However, despite the work's merits, the manuscript in its present form is confusing and may be misleading for the reader.

The work is initially presented in the simple summary, abstract and introduction in the context of strategies to combat antibiotic resistance. Although, all work is directed towards the possible anti-inflammatory effects of anemoside B4.

Answer: Thanks for your suggestion. The content has been revised. The main purpose of this study was to explore the changes in metabolites in clinical mastitis cows treated by anemoside B4.

In a study on clinical mastitis in cattle, the clinical diagnosis and SCC only establish the presumptive diagnosis. Laboratory confirmation of the agents involved is mandatory. Thus, the manuscript must be rewritten and resubmitted. With special attention to the study framework, clear definition of objectives, objective-oriented discussion (the authors already have an adequate discussion directed towards laboratory results) and respective conclusions

Answer: Thanks. Animals diagnosed with mastitis were also confirmed by laboratory tests, which had been published in our previous studies . Our previous studies showed that the bacteria that cause mastitis included Escherichia coli, cocci and mycoplasma, mainly in mixed infection. The main types of cocci were Staphylococcus epidermidis, Staphylococcus chromogenes and Staphylococcus haemolyticus. We have added relevant content in the manuscript.

Reviewer 3 Report

Dear Authors,

I hope this letter finds you well. I have had the opportunity to review your article titled "Effects of anemoside B4 on plasma metabolites in cows with 2 clinical mastitis 3," which has been submitted for consideration . I would like to provide feedback on an important aspect that I believe needs to be addressed regarding the control group and the absence of clear selection criteria for the cows involved, such as age, previous illnesses, somatic cell count, and so on. Furthermore, I would like to express my curiosity regarding the exclusion of animals that received the medication but did not show improvement. These aspects are crucial for the assessment of the drug's effectiveness.

Firstly, it is essential to include a control group that receives a placebo and fulfills the criteria of having the disease. This control group is crucial for evaluating the true effect of the medication being studied. By comparing the results of the treatment group to those of the control group, one can determine whether the observed effects are genuinely attributable to the medication or are possibly due to other factors. I recommend incorporating a control group that mirrors the characteristics of the experimental group in terms of disease presence and administration of a placebo.

Secondly, the eligibility criteria for the cows involved in the study are not adequately described. Specific factors such as age, previous illnesses, somatic cell count, and other relevant indicators play a significant role in ensuring the homogeneity of the study population. It is important to establish transparent and well-defined criteria to ensure that the selected cows accurately represent the target population. This will enhance the validity and generalizability of the study findings. I recommend providing a comprehensive description of the selection criteria used for the cows, ensuring that all relevant factors are considered.

Additionally, I would like to express my curiosity regarding the exclusion of animals that received the medication but did not show improvement. While it is understandable to focus on positive outcomes, the inclusion of non-responsive animals is crucial for a comprehensive assessment of the drug's effectiveness. It would be valuable to provide a rationale for their exclusion and to discuss any potential implications for the interpretation of the study results.

Addressing these concerns will significantly strengthen the quality and impact of your research. I kindly request that you carefully reconsider the design of the control group, provide explicit selection criteria for the cows, and offer an explanation for the exclusion of non-responsive animals. These revisions will enhance the validity of your findings and contribute to the overall scientific knowledge in this field.

I appreciate the effort and dedication you have invested in this research. I look forward to reviewing the revised manuscript and commend you for your commitment to addressing these concerns.

If you require any further clarification or have any questions regarding the feedback provided, please do not hesitate to reach out to me. Thank you for your attention to these matters.

Yours sincerely,

Author Response

Comments and Suggestions for Authors

Firstly, it is essential to include a control group that receives a placebo and fulfills the criteria of having the disease. This control group is crucial for evaluating the true effect of the medication being studied. By comparing the results of the treatment group to those of the control group, one can determine whether the observed effects are genuinely attributable to the medication or are possibly due to other factors. I recommend incorporating a control group that mirrors the characteristics of the experimental group in terms of disease presence and administration of a placebo.

Answer: Thanks for your suggestion, but we don't think having a control group in this case is a good choice, not only because of animal welfare but also because of financial loss. We treated diseased animals and the control group consisted of healthy animals. Our study design is close to reality and can provide useful information and meet our study objectives.

Secondly, the eligibility criteria for the cows involved in the study are not adequately described. Specific factors such as age, previous illnesses, somatic cell count, and other relevant indicators play a significant role in ensuring the homogeneity of the study population. It is important to establish transparent and well-defined criteria to ensure that the selected cows accurately represent the target population. This will enhance the validity and generalizability of the study findings. I recommend providing a comprehensive description of the selection criteria used for the cows, ensuring that all relevant factors are considered.

Answer: Animals in both groups were similar in age, weight, parity and body condition score to minimize the influence of other factors. Animals in the diseased group got mastitis and animals in the control group were all healthy as diagnosed by a vet. Details can be found in 2.3 Experimental design, cows and management, ‘Cows with CM were diagnosed by a veterinarian, based on the obvious symptoms of redness in udders, clots in milk, discoloring, and the SCCs in milk was more than 500,000 cells/ ml. Mastitis-affected Holstein cows (milk SCCs = 2,844,100 ± 61.64 cells/ ml; milk yield = 25.68 ± 2.13 kg/day; parity = 2.65 ± 0.27) were selected… 9 disease-free dairy cows from the same breed and with a similar body condition (milk SCCs = 184,700 ± 4.13 cells/ ml; milk yield = 37.97 ± 3.37 kg/day; parity 2.48 ± 0.34; free of diseases) were selected as controls.’

Additionally, I would like to express my curiosity regarding the exclusion of animals that received the medication but did not show improvement. While it is understandable to focus on positive outcomes, the inclusion of non-responsive animals is crucial for a comprehensive assessment of the drug's effectiveness. It would be valuable to provide a rationale for their exclusion and to discuss any potential implications for the interpretation of the study results. Addressing these concerns will significantly strengthen the quality and impact of your research. I kindly request that you carefully reconsider the design of the control group, provide explicit selection criteria for the cows, and offer an explanation for the exclusion of non-responsive animals. These revisions will enhance the validity of your findings and contribute to the overall scientific knowledge in this field.

Answer: Thank you so much for your sincere suggestions. Our previous study confirmed that anemoside B4 were effective against Staphylococcus epidermidis, Staphylococcus chromogenes, Staphylococcus haemolyticus and Escherichia coli for cows with CM, but not against Staphylococcus aureus. We therefore speculated that these uncured animals might be primarily infected with Staphylococcus aureus. We were interested in studying the effects of anemoside B4 on metabolites in cows cured from CM, so those uncured cows were excluded. We have explained and discussed in the Discussion section.

Round 2

Reviewer 1 Report

My comments were addressed appropriately by the authors.

Author Response

Thank you very much for your help. Your kind suggestions made our manuscript improved a lot.

Reviewer 2 Report

The authors made remarkably significant improvements over the previous manuscript. The work "Effects of anemoside B4 on plasma metabolites in cows with clinical mastitis" the potential to be published after additional changes.

Line 48-49: This sentence needs to be modified. Without questioning the extraordinary usefulness and potential of Traditional Chinese Medicine, obviously in science any therapy or medicine has potentially beneficial and non-beneficial elements.

Therefore, generic sentences are reckless and can be misleading for the reader, so they should be avoided in a scientific article. This article explores very well the potential of anemoside B4 on plasma metabolites in cows with clinical mastitis. This is what needs to be highlighted at this point. Its framing within the scope of the strategies to combat antibiotic resistance requires further investigation.

Line 91 (or M&M:  Experimental design, cows and management)

Regarding my previous comment: "In a study on clinical mastitis in cattle, the clinical diagnosis and SCC only establish the presumptive diagnosis. Laboratory confirmation of the agents involved is mandatory. Thus, the manuscript must be rewritten and resubmitted. With special attention to the study framework, clear definition of objectives, objective-oriented discussion (the authors already have an adequate discussion directed towards laboratory results) and respective conclusions".

The information available regarding the authors' previous study (reference 18 of this article) helps to clarify  partially this issue. But it is not obvious to the reader. So, the methodology must be reformulated.

In summary, it is important to clarify whether the experiment is the same as in reference 18. And include it in the M&M. Or in the absence of laboratory confirmation in this particular assay, this constitutes a serious flaw in the methodology.

Author Response

Thank you very much. Our writing will be more rigorous. It has been revised, ‘Many traditional Chinese medicine and its extracts have been developed into medicines for the treatment of such diseases and have the potential to address both problems.’.

We have revised the M&M, hope it meets your expectation. Thank you again for your professional suggestions, which make our research become better and better. 

Reviewer 3 Report

Dear Authors,

Re: Review of the Paper "Effects of anemoside B4 on plasma metabolites in cows with 2 clinical mastitis"

I hope this letter finds you well. I am writing to provide you with feedback on the revised version of your manuscript, titled "Effects of anemoside B4 on plasma metabolites in cows with 2 clinical mastitis," which was submitted to Veterinary Science. I have carefully reviewed the manuscript and I am pleased to inform you that you have adequately addressed the suggestions and recommendations put forth by the reviewers. I commend you for the thoroughness and diligence with which you have revised your work.

In response to the reviewers' comments, you have made significant improvements to the clarity and organization of the manuscript. The revised introduction now provides a comprehensive overview of the research area and clearly states the objectives of your study. The methodology section has been expanded and includes additional details that enhance the reproducibility of your experiments. Moreover, the results section now presents the findings in a logical and coherent manner, supported by appropriate statistical analysis.

Based on the revisions made, I believe that your manuscript is now suitable for publication in Veterinary Science

Author Response

We do appreciate your help in the process of reviewing our manuscript. We thank all the reviewers for their kind suggestions and their time. We learnt a lot from the whole process and we do believe that our work is important in this field and will be very useful for other researchers.